# Simulated Warming Reduces Biomass Accumulation in *Zizania caduciflor* and *Sparganium stoloniferum*

**DOI:** 10.3390/plants14101414

**Published:** 2025-05-09

**Authors:** Tingfeng Wang, Junbao Yu, Yun Zhang, Kun Tian, Xiangyu Zhu, Mei Sun, Zhenya Liu

**Affiliations:** 1Yunnan Key Laboratory of Plateau Wetland Concervation, Restoration and Ecological Services, Southwest Forestry University, Kunming 650224, China; wang12112010@163.com (T.W.); yu.junbao@gmail.com (J.Y.); zhangyuncool@163.com (Y.Z.); tlkunp@126.com (K.T.); m15808639787@163.com (X.Z.); 2National Plateau Wetland Research Center, Southwest Forestry University, Kunming 650224, China; 3National Wetland Ecosystem Fixed Research Station of Yunnan Dianchi, Jinning 650600, China

**Keywords:** climate change, northwestern Yunnan, plateau wetlands, emergent plant, plant functional traits

## Abstract

Climate change, represented by global warming, significantly affects the structure and function of alpine wetland ecosystems. Investigating the response strategies of alpine wetland plants to temperature changes is fundamental to understanding how alpine wetlands cope with global warming. This study, conducted at the typical alpine wetland Napahai, uses the latest predictions from the Intergovernmental Panel on Climate Change (IPCC) and employs open–top chamber warming experiments (OTCs) to study the responses of typical alpine wetland plants, *Zizania caduciflor* and *Sparganium stoloniferum*, to simulated warming. The results indicate that simulated warming significantly reduced the photosynthetic capacity of *Z. caduciflor*, and obviously decreased the biomass accumulation of both *Z. caduciflor* and *S. stoloniferum* (*p* < 0.05). The mean annual temperature (MAT) and annual maximum temperature (max) are the primary temperature factors affecting the photosynthetic and biomass parameters. Specifically, the net photosynthetic rate, stomatal conductance, transpiration rate, the aboveground, underground, and total biomasses, and the nitrogen contents of aboveground and underground buds of *Z. caduciflor* all showed significant negative correlations with MAT and max (*p* < 0.05). The parameters of *S. stoloniferum* mainly showed significant correlations with max, with its underground biomass, total biomass, and root nitrogen content all showing significant negative correlations with max, while its fibrous root carbon content and underground bud phosphorus content showed significant positive correlations with max (*p* < 0.05). The results are consistent with previous studies in high–altitude regions, indicating that warming reduces the photosynthetic capacity and biomass accumulation of alpine wetland plants, a trend that is widespread and will lead to a decline in the productivity of alpine wetlands and changes in vegetation composition. The study can provide a case for understanding the response strategies of alpine wetlands in the context of climate change.

## 1. Introduction

Climate change, with global warming as a prominent manifestation, is currently one of the most challenging environmental issues faced by humanity [1]. According to the Fifth Assessment Report on Climate Change by the Intergovernmental Panel on Climate Change (IPCC), atmospheric CO_2_ concentrations will reach 730–1000 ppm by 2100, with temperatures increasing by 1–3.7 °C in tandem [2]. Global warming significantly impacts the structure and function of wetland ecosystems, with the effects being particularly pronounced in high–altitude regions [3,4,5,6]. Highland wetlands in China account for 48.22% of the country’s total wetland area [7,8] and are an essential component of China’s “two screens and three belts” ecological security pattern [9,10,11]. The northwestern region of Yunnan, located on the southeastern edge of the Tibetan Plateau, possesses typical Asian subtropical Himalayan highland wetlands, with significant altitudinal differences and distinct environmental gradients [7,12]. Influenced by differential uplift due to neotectonic movements, fault subsidence, glacial erosion, and fluvial modification, this area has developed numerous small individual wetlands with high spatial heterogeneity and no interconnected waterways, forming unique closed and semi–closed wetland types. The wetland ecosystems in this region are extremely fragile [12]. The lakeshore zone is a primary component of the alpine wetland ecosystems in northwestern Yunnan, and plants, as the basis for the structure and function of the lakeshore zone, are highly sensitive to changes in climatic conditions [13,14]. Researching the ecological response strategies of lakeshore zone plants to climate warming in northwestern Yunnan is an important aspect for people to effectively understand the dynamics of alpine wetlands in northwestern Yunnan and to scientifically respond to climate change. It is also a hot topic of interest both domestically and internationally.

Plant functional traits can effectively modulate plant responses and adaptations to environmental changes, and are therefore often used to discuss patterns and mechanisms of plant environmental adaptation [15,16,17,18]. Research on plant adaptability emphasizes the quantitative determination of plant functional trait syndromes [19], one important aspect of which is to understand the trends, extents, and quantitative relationships of traits with environmental changes, and to elucidate the reasons for different traits working together to adapt to environmental changes [19]. Temperature is one of the most important environmental factors determining plant physiological functions, adaptive strategies, and distribution ranges [20]. Plants exhibit varying degrees of adaptation in their morphological structure and physiological functional traits in response to temperature changes [19,20]. Within the optimal temperature range for plants, the photosynthetic rate of most plants increases with rising temperatures, but as temperatures continue to rise beyond the optimal range, the photosynthetic rate declines [21,22]; environmental temperature affects plant growth, reproduction, and material accumulation by impacting photosynthetic functions [23,24]; high temperatures are a major abiotic factor limiting plant growth and production [25]; on a global scale, the annual mean temperature is significantly negatively correlated with plant nitrogen (N) and phosphorus (P) content [26]; and Roche et al. found in a study of 10 plant species across 8 sites in the southern Mediterranean that the annual mean minimum temperature is significantly negatively correlated with leaf biomass accumulation [27]. The significant correlation between plant functional traits and temperature fully reflects the response strategies of plants to temperature changes. However, these correlations have been less studied in alpine wetland plants, and the main temperature factors affecting plant functional traits and adaptability are not clear. There are also few reports on how climatic effects manifest in alpine wetland plant communities and their environmental responses [12].

*Zizania latifolia* and *Sparganium stoloniferum* are dominant emergent plants in the lakeshore zones of Northwest Yunnan, playing a significant role in the ecological structure and functions of these zones and being highly sensitive to environmental changes [28]. The presence of these two species is an important indicator of well–functioning wetland ecosystems in the wetlands of the area [29,30,31]. This study focuses on the ecological response of alpine wetland plants to climate change, selecting the typical alpine wetland Napahai in northwestern Yunnan as the research site, and targeting the large emergent plants *Z. caduciflor* and *S. stoloniferum* in the lakeshore zone. Based on the IPCC’s predictions for future atmospheric warming, an open–top warming control experiment is conducted using an artificial warming system. By observing the changes in the photosynthetic physiological parameters, biomass, and nutrient element content of *Z. caduciflor* and *S. stoloniferum* under different growth temperatures, the study explores the adaptive changes of plant functional traits to warming and the main temperature factors affecting plant functional traits. This research provides a case study for revealing the physiological adaptation strategies of alpine wetland plants to global warming.

## 2. Results

### 2.1. Differences in Functional Traits of Z. caduciflora and S. stoloniferum Under Three Different Growth Temperatures

Compared with the control group (CK), the net photosynthetic rate (*P_n_*) and stomatal conductance (*G_s_*) of *Z. caduciflor* under simulated warming treatment were significantly reduced, but there were no significant differences among the two parameters between the 2.0 ± 0.5 °C (ET–2) and 4.0 ± 0.5 °C (ET–4) warming treatments; the traspiration rate (*T_r_*) of *Z. caduciflor* under the ET–2 simulated warming treatment was also significantly reduced, while the photosynthetic parameters of *S. stoloniferum* showed no significant differences among CK, ET–2, and ET–4 (Figure 1). Additionally, the photosynthetic parameters of *S. stoloniferum* were mostly numerically higher than those of *Z. caduciflor* overall (Figure 1).

Simulated warming tends to reduce the biomass accumulation of both plants. Compared with CK, the aboveground biomass (Bio_above_) and total biomass (Bio_total_) of *Z. caduciflora* showed no significant differences under the ET–2 treatment, but significantly decreased under the ET–4 treatment; the interval biomass (Bio_inter_) and underground biomass (Bio_under_) significantly decreased under both the ET–2 and ET–4 treatments, but there were no significant differences between these two warming treatments (Figure 1). From the CK to the ET–2, and then to the ET–4, with the increase in temperature, the underground bud biomass (Bio_bud_) of *Z. caduciflor* showed a significant decreasing trend, while the biomass changes in the fibrous roots were not significant (Figure 1).

The Bio_above_, Bio_inter_, and Bio_total_ of *S. stoloniferum* showed no significant differences between the CK and ET–4 groups, but were significantly lower in the ET–2 group, indicating a trend of first decreasing and then increasing with the rise in temperature (Figure 1). Meanwhile, compared with CK, the Bio_bud_ and Bio_under_ of this species were significantly reduced in both the ET–2 and ET–4 treatments. At the same time, the biomass of each part of *Z. caduciflor* was significantly higher than that of *S. stoloniferum* (Figure 1).

Compared with the CK group, the simulated warming had a smaller impact on the aboveground carbon content (C_above_) of both plant species, and the changes did not reach a significant level (*p* > 0.05; Figure 2). Among the belowground components, compared with the CK group, the fibrous root carbon content (C_fib_) of *Z. caduciflor* significantly decreased under the ET–4 condition, while the underground bud carbon content (C_bud_) was significantly higher under both warming conditions (*p* < 0.05; Figure 2). For *S. stoloniferum*, C_fib_ was significantly higher under the ET–2 condition, but returned to a level comparable to the CK group under the ET–4 condition, and there were no significant differences in the carbon content of other belowground components among the three groups (*p* > 0.05; Figure 2).

Compared with the CK group, the aboveground nitrogen content (N_above_) of *Z. caduciflor* was significantly lower under the ET–2 and ET–4 treatments (*p* < 0.05), with no significant differences between the ET–2 and ET–4 treatments (Figure 2); the N_above_ of *S. stoloniferum* significantly decreased under the ET–2 treatment (*p* < 0.05), and then returned to the original level under the ET–4 treatment (*p* > 0.05; Figure 2). Among the belowground components, compared with the CK group, the interval nitrogen content (N_inter_) and fibrous root nitrogen content (N_fib_) of *Z. caduciflor* showed no significant differences among the three groups (*p* > 0.05), while the underground bud nitrogen content (N_bud_) significantly decreased under the ET–4 treatment (*p* < 0.05; Figure 2). For *S. stoloniferum*, N_inter_ showed no significant differences among the three groups (*p* > 0.05); N_fib_ significantly increased under the ET–2 treatment (*p* < 0.05), while the root nitrogen content (N_root_) significantly decreased under the ET–2 treatment (*p* < 0.05), and both returned to the CK level under the ET–4 treatment (*p* > 0.05); its N_fib_ showed no significant difference between the CK and ET–2 (*p* > 0.05) treatment, but significantly increased under the ET–4 treatment (*p* < 0.05; Figure 2).

Compared with the CK group, there were no significant differences in phosphorus content in various parts of *Z. caduciflor* among the three groups (*p* > 0.05). The aboveground phosphorus content (P_above_) of *S. stoloniferum* showed no significant differences among the three groups (*p* > 0.05), while the phosphorus content in each component of the belowground parts tended to increase under warming treatments (Figure 2). Specifically, the interval phosphorus content (P_inter_) of *S. stoloniferum* significantly increased under the ET–4 condition; its fibrous root phosphorus content (P_fib_) significantly increased under the ET–2 condition; and both the root phosphorus content (P_root_) and the underground bud phosphorus content (P_bud_) were significantly higher under both the ET–2 and ET–4 conditions (*p* < 0.05; Figure 2). Additionally, overall, the phosphorus content in each component of *S. stoloniferum* was higher than that in the corresponding components of *Z. caduciflor* (Figure 2).

### 2.2. The Relationship Between Z. caduciflora and S. stoloniferum Functional Traits and Environmental Factors

The stepwise regression models based on the functional traits of *Z. caduciflor* and *S. stoloniferum* and temperature factors indicate that the mean annual temperature (MAT) and the annual maximum temperature (max) are the main temperature factors affecting the trait variations in both species (Table 1 and Table 2). Specifically, the Bio_above_, C_above_, C_fib_, and N_bud_ of *Z. caduciflora* are mainly influenced by the MAT and max. The *P_n_*, *G_s_*, *T_r_*, C_bud_, and N_above_ are primarily affected by the max. The Bio_inter_, Bio_fib_, Bio_bud_, Bio_under_, and Bio_total_ are mainly influenced by the MAT. However, its intercellular CO_2_ concentration (*C_i_*), Bio_fib_, C_inter_, N_inter_, N_fib_, and phosphorus content in each component are less affected by temperature factors, with non–existent or non–significant stepwise regression models (Table 1).

For *S. stoloniferum,* most parameters such as the *P_n_* and *G_s_*, all biomass parameters of each component, the nitrogen content in each component, and the phosphorus content in each component are mainly influenced by the MAT and max. The C_fib_ is primarily affected by the max. The *T_r_*, *C_i_*, Bio_above_, C_above_, C_inter_, C_root_, C_bud_, N_inter_, N_root_, and P_above_ are less affected by temperature factors, with non–existent or non–significant models (Table 2).

The results of the redundancy analysis (RDA) based on the functional traits and temperature factors of *Z. caduciflora* show that the temperature factors account for a total cumulative explanation of 57.09% of the traits, with the first and second ordination axes explaining 42.93% and 14.16%, respectively (Figure 3a). The photosynthetic parameters, biomass parameters, and most elemental content parameters of *Z. caduciflora* are mainly distributed on the positive side of the first principal axis, while the C_bud_ and P_inter_ are mainly distributed on the negative side of the first principal axis (Figure 3a). The C_above_, C_fib_ N_inter_, and T_r_ are mainly distributed on the second principal axis; the temperature factor parameters including the MAT, max, min, dat, nat, and sat are mainly distributed on the negative side of the first principal axis (Figure 3a). According to the distribution of traits and temperature factors, the photosynthetic parameters, biomass, and most elemental content parameters of *Z. caduciflora* are significantly influenced by temperature factors.

The RDA results based on the functional traits and temperature factors of *S. stoloniferum* show that the temperature factors account for a total cumulative explanation of 62.12% of the traits, with the first two ordination axes explaining 44.55% and 17.57%, respectively (Figure 3b). The *P_n_*, *G_s_*, and *C_i_* of *S. stoloniferum* are mainly distributed on the positive side of the first principal axis, while the biomass parameters of each part are mainly distributed on the negative side of the first principal axis, and the elemental content of each part is distributed on both the first and second principal axes (Figure 3b). Among the temperature factors, the max is mainly distributed on the positive side of the first principal axis, while the other temperature factors are mainly distributed on the positive side of the second principal axis (Figure 3b). According to the distribution of traits and temperature factors, the photosynthetic traits besides *T_r_* and the biomass of *S. stoloniferum* are mainly related to the max, while its *T_r_* and elemental content are more influenced by other temperature factors.

Combining the results of the stepwise regression model and RDA, the *C_i_*, Bio_fib_, C_inter_, N_inter_, N_fib_, and phosphorus content in each component of *Z. caduciflora* is less affected by temperature factors, while other traits are mainly influenced by the MAT and max (Table 1, Figure 3a). According to the bivariate correlation analysis between the functional traits of *Z. caduciflora* and these two temperature factors (Table 3), among the photosynthetic parameters, the *P_n_* and *G_s_* have significant negative correlations with the MAT and max, and the *T_r_* also has a significant negative correlation with the max (Table 3). Among the biomass parameters, the Bio_inter_, Bio_bud_, Bio_under_, and Bio_total_ have significant negative correlations with the MAT and max, and the Bio_above_ also has a significant negative correlation with the MAT (Table 3). Among the elemental contents, the C_bud_ has significant positive correlations with the MAT and max; the N_above_ shows significant negative correlations with the MAT and max; and the N_bud_ also has a significant negative correlation with the MAT (Table 3).

**Table 3 plants-14-01414-t003:** Pearson correlation between functional traits and temperature factors of *Z. caduciflora* and *S. stoloniferum*.

Functional Traits	Mean Annual Temperature/MAT		Annual Maximum Temperature/Max	
	*Z. caduciflora*	*S. stoloniferum*	*Z. caduciflora*	*S. stoloniferum*
net photosynthetic rate	**−0.852 ****	0.315	**−0.937 *****	0.515
stomatal conductance	**−0.828 ****	0.278	**−0.937 *****	0.490
transpiration rate	−0.586	–	**−0.743 ***	–
intercellular CO_2_ concentration	–	–	–	–
aboveground biomass	**−0.727 ***	–	−0.523	–
interval biomass	**−0.908 *****	−0.638	**−0.832 ****	**−0.826 ****
fibrous root biomass	–	−0.437	–	**−0.705 ***
root biomass	–	−0.406	–	−0.641
bud biomass	**−0.960 *****	**−0.878 ****	**−0.888 *****	**−0.967 ****
underground biomass	**−0.893 *****	−0.521	**−0.820 ****	**−0.761 ***
total biomass	**−0.862 ****	−0.456	**−0.724 ***	**−0.710 ***
aboveground carbon comtent	−0.043	–	0.184	–
interval carbon comtent	–	–	–	–
fibrous root carbon comtent	−0.558	**0.676 ***	−0.259	**0.821 ****
root carbon comtent	–	–	–	–
bud carbon comtent	**0.898 *****	–	**0.989 *****	–
aboveground nitrogen comtent	**−0.822 ****	−0.229	**−0.904 *****	−0.509
interval nitrogen comtent	–	−0.570	–	−0.369
fibrous root nitrogen comtent	–	−0.117	–	0.226
root nitrogen comtent	–	−0.479	–	**−0.679 ***
bud nitrogen comtent	**−0.828 ****	**0.877 ****	−0.565	0.666
aboveground phosphorus comtent	–	–	–	–
interval phosphorus comtent	–	–	–	–
fibrous root phosphorus comtent	–	0.201	–	0.493
root phosphorus comtent	–	–	–	–
bud phosphorus comtent	–	0.444	–	**0.740 ***

In this context, the underground part of the *Z. caduciflor* does not include the roots. Bold fonts indicate a significance level of less than 0.05. * *p* < 0.05; ** *p* < 0.01; *** *p* < 0.001.

The *T_r_*, *C_i_*, Bio_above_, C_above_, C_inter_, C_root_, C_bud_, P_above_, P_inter_, and P_root_ of *S. stoloniferum* are less affected by temperature factors, while other traits are more significantly influenced by the max and relatively less influenced by the MAT (Table 2, Figure 3b). The photosynthetic parameters of *S. stoloniferum* do not show significant correlations with the MAT or max (Table 3). Among the biomass parameters, except for root biomass, other biomass parameters have significant negative correlations with the max, and Bio_bud_ also has a significant negative correlation with the MAT (Table 3). Among the elemental content parameters, C_fib_ has significant positive correlations with both the MAT and max; N_bud_ has a significant positive correlation with the MAT; and P_bud_ has a significant positive correlation with the max (Table 3).

## 3. Materials and Methods

### 3.1. Overview of the Study Area

Our study site is located in the Napahai Provincial Nature Reserve. Napahai Wetland (99°37′11″–99°40′20″ E, 27°48′56″–27°54′28″ N) is situated in Shangri–La City, Diqing Tibetan Autonomous Prefecture, Yunnan Province, in the middle section of the Hengduan Mountains, and is a seasonal lake wetland at low latitude and high altitude. Influenced by differential uplift due to neotectonic movements, fault subsidence, glacial erosion, and fluvial modification, the Hengduan Mountain area generally features small individual wetlands with high spatial heterogeneity and no interconnected waterways, forming unique closed and semi–closed wetland types, and Napahai Wetland also conforms to these characteristics. Due to its enclosed nature, small wetland area, and the characteristic of water source replenishment mainly relying on rainfall and snowmelt, the Napahai Wetland ecosystem is relatively fragile and highly sensitive to human activities and climate change.

The Shangri–La region, where Napahai Wetland is located, belongs to the western type of the cold temperate plateau monsoon climate zone. Influenced by the north–south arranged mountains and atmospheric circulation, the southerly and southwesterly winds prevail throughout the year. Due to its location in the southeastern extension of the Tibetan Plateau, it has distinct plateau climatic characteristics. The climatic features of Napahai include strong solar radiation, with an annual average of 2180 h of sunshine; a small annual temperature range, a large diurnal temperature range, long winters without summers, and short springs and autumns. The average annual temperature is 5.4 °C, the average temperature in the hottest month of July is 13.2 °C, and the average temperature in the coldest month of January is −3.7 °C, using a base temperature of 10 °C, and the accumulated temperature (or thermal time or Growing Degree Days, GDDs) reached 1392.8 °C. Napahai is situated in the transitional zone between rainy and less rainy areas, with an annual average precipitation of 620 mm, an annual frost period of about 125 days, and snow from September to May of the following year.

Napahai’s unique geographical environment has given rise to rich biodiversity, which is conducive to the differentiation of endemic species and the preservation of ancient species [32]. The wetland is rich in plant species, including two major categories: hydrophytes and emergent plants. Among them, submerged plants include *Hydrailla verticillata* and *Myriophyllum spicatum*; floating–leaf plants mainly consist of *Potamogeton maackianus*; and emergent plants include *Scirpus validus*, *Zizania caduciflora*, *Sparganium stoloniferum*, *Hippuris vulgaris*, and others.

### 3.2. Experimental Design

#### 3.2.1. Construction of Artificial Simulated Warming Chambers

Open–top chambers (OTCs) were constructed in April 2014. Nine in situ research units with a bottom diameter of 2.4 m were established in an area similar to the Napahai wetland environment, with a 3 m interval between each research unit and connected with PVC pipes to ensure that the water depth in each research unit was consistent with the natural flooding depth in the field (approximately 18 cm). According to the IPCC’s assessment report for the end of the 21st century, the global near–surface atmospheric temperature is expected to rise by 0.3–1.7 °C under the RCP2.6 scenario and by 2.6–4.8 °C under the RCP8.5 scenario [2]. Based on this, we divided the research units into three groups, with three replicates per group (Figure 4). One group was set as the control group (CK), consistent with the surrounding environmental conditions, and the other two groups were set as experimental groups. Open–top simulated warming chambers were constructed on the ground with sunlight board materials, reaching a height of 2.4 m. By controlling the opening size of the chambers, different degrees of indoor temperature increase were achieved. One group of chambers had an opening size set to achieve an atmospheric warming of 2.0 ± 0.5 °C (ET–2), and the other group had an opening size set to achieve an atmospheric warming of 4.0 ± 0.5 °C (ET–4).

In the center of each of the 9 differently treated research units, 1 m above the water surface, a TP–2200 temperature real–time recorder was placed. Each recorder automatically logged the atmospheric temperature of each research unit 24 times a day at a recording frequency of 1 time per hour. After the experiment concluded, the data from the temperature real–time recorders were exported, and the following data were organized and calculated for each group: the mean annual temperature (MAT, °C), annual maximum temperature (max, °C), annual minimum temperature (min, °C), seasonal average temperature (sat, °C), daytime accumulated temperature (dat, °C), and nighttime accumulated temperature (nat, °C). According to the analysis results of the full–year simulated warming in 2018, the average atmospheric temperature increase in the ET–2 and ET–4 groups was 2.17 °C and 3.76 °C, respectively (Table 4), indicating that the OTCs’ simulated warming system used in this study provided a stable warming effect.

**Table 4 plants-14-01414-t004:** Temperature variable values under different temperature conditions.

Process	Temperature Variable					
	MAT	Max	Min	Sat	Dat	Nat
CK	8.83	37.69	−22.38	15.12	2572.79	281.66
ET–2	11.00	46.88	−19.13	17.76	3319.56	470.34
ET–4	12.59	47.50	−16.75	19.09	3877.70	562.04

CK, ambient temperature (control group); ET–2, (2.0 ± 0.5) °C warming; ET–4, (4.0 ± 0.5) °C warming. MAT: mean annual temperature, max: annual maximum temperature, min: annual minimum temperature, sat: seasonal average temperature, dat: daytime accumulated temperature, nat: nighttime accumulated temperature.

#### 3.2.2. Plant Transplantation

Based on preliminary surveys of dominant plants in the Napahai lakeshore zone, this study selected *Z. caduciflor* and *S. stoloniferum* as the subjects of research. In March–April 2018, healthy and similarly vigorous clones of these two plant species were excavated from the Napahai lakeshore zone. Soil from the Napahai lakeshore zone at a depth of 30 cm was collected as the cultivation substrate, and individual plants were transplanted into experimental pots with a diameter of 35 cm and a height of 40 cm, with an equal amount of soil in each pot. After a 15–day acclimation period under natural conditions, the transplanted plants were randomly divided into three groups (three pots per group) and then placed into different research units (Figure 4). To exclude edge effects and ensure uniform light conditions, the pots were randomly placed at the midpoint between the inner edge and the center of the growth chamber. The water depth in each research unit was based on the native waterlogged environment (20–30 cm) (Figure 4). Subsequent daily management of the plants was intensified to ensure the survival of the transplanted plants.

### 3.3. Measurement of Trait Parameters

#### 3.3.1. Measurement of Photosynthetic Parameters

During the peak growth period of the plants in 2018 (July–August), from 9:00 to 12:00 on sunny mornings, the net photosynthetic rate, stomatal conductance, intercellular CO_2_ concentration, and transpiration rate of plants in different growth chambers were recorded in situ using the Li-6800XT portable photosynthesis system (LI-6800, LI-COR, 4647 Superior Street, Lincoln, NE 68504, USA). In each growth chamber, three mature plants were selected as subjects, and two healthy, fully expanded leaves per plant were chosen for measurement. During the measurement, the internal light intensity of the leaf chamber was set to 1500 μmol·m^−2^·s^−1^, the leaf temperature was maintained at 22–24 °C, the flow rate was set to 500 μmol·s^−1^, and the reference chamber CO_2_ concentration was set to 425 μmol·mol^−1^.

#### 3.3.2. Determination of Biomass and Elemental Content

The entire plants of *Z. caduciflor* and *S. stoloniferum*, from which photosynthetic parameters had been measured, were carefully removed, rinsed with running water, marked, and brought back to the laboratory. In the laboratory, the aboveground and belowground parts were separated, dividing the entire plant into aboveground and belowground sections. The belowground part was further divided into interval, fibrous root, root, and underground buds; in this context, the underground part of the *Z. caduciflor* does not include the roots. After weighing the fresh weight of each part, they were individually placed into numbered kraft envelopes and dried in a 75 °C oven for at least 48 h until they reached a constant weight. This process yielded the biomass of each part, including the aboveground biomass, interval biomass, fibrous root biomass, root biomass, and underground bud biomass. The underground biomass was calculated as the sum of the latter four parts, and the total biomass was determined by adding the aboveground biomass to the underground biomass.

After weighing the biomass, the plant parts were ground separately using a plant grinder and mortar, and then sifted through a sieve with a mesh size of 0.25 mm. The sifted powder was packaged in labeled sealable bags and stored in a refrigerator at 4 °C for future use. Three milligrams of the ground plant samples was taken, treated with acid, dried, and wrapped in tin foil for analysis using the vario TOC select total organic carbon analyzer from Elementar, company is located in Hanau, a city near Frankfurt, Germany, to determine the total carbon mass fraction of each plant part, denoted as the total carbon content. Next, 0.2 g of the plant powder was digested using the concentrated sulfuric acid (H_2_SO_4_) and hydrogen peroxide (H_2_O_2_) digestion method, after which the solution was made up to volume and filtered. The filtrate was then used to measure and calculate the total nitrogen and phosphorus mass fractions of the plants with an AA3 continuous flow analyzer, denoted as the total nitrogen content and total phosphorus content, respectively. The plant functional traits measured in this study and their abbreviations are shown in Table 5.

**Table 5 plants-14-01414-t005:** Measured plant functional traits and abbreviations.

Functional Traits		Abbreviations (Unit)
Photosynthetic parameters	net photosynthetic rate	*P_n_* (μmol·m^−2^·s^−1^)
	stomatal conductance	*G_s_* (mol·m^−2^·s^−1^)
	transpiration rate	*T_r_* (μmol·mol^−1^)
	intercellular CO_2_ concentration	*C_i_* (mmol·m^−2^·s^−1^)
Biomass	aboveground biomass	Bio_above_ (g·m^−2^)
	interval biomass	Bio_inter_ (g·m^−2^)
	fibrous root biomass	Bio_fib_ (g·m^−2^)
	root biomass	Bio_root_ (g·m^−2^)
	bud biomass	Bio_bud_ (g·m^−2^)
	underground biomass	Bio_under_ (g·m^−2^)
	total biomass	Bio_total_ (g·m^−2^)
Elemental content	aboveground carbon comtent	C_above_ (g·kg^−1^)
	interval carbon comtent	C_inter_ (g·kg^−1^)
	fibrous root carbon comtent	C_fib_ (g·kg^−1^)
	root carbon comtent	C_root_ (g·kg^−1^)
	bud carbon comtent	C_bud_ (g·kg^−1^)
	aboveground nitrogen comtent	N_above_ (g·kg^−1^)
	interval nitrogen comtent	N_inter_ (g·kg^−1^)
	fibrous root nitrogen comtent	N_fib_ (g·kg^−1^)
	root nitrogen comtent	N_root_ (g·kg^−1^)
	bud nitrogen comtent	N_bud_ (g·kg^−1^)
	aboveground phosphorus comtent	P_above_ (g·kg^−1^)
	interval phosphorus comtent	P_inter_ (g·kg^−1^)
	fibrous root phosphorus comtent	P_fib_ (g·kg^−1^)
	root phosphorus comtent	P_root_ (g·kg^−1^)
	bud phosphorus comtent	P_bud_ (g·kg^−1^)

### 3.4. Data Analysis

This study employed SPSS 25.0 software “www.ibm.com/products/spss-statistics (accessed on 10 August 2024)” to test the homogeneity of variance of the photosynthetic traits, biomass, and elemental mass fractions using Levene’s test. Then, to conduct a one–way ANOVA to test for differences in the functional traits between the three temperature treatment groups of *Z. caduciflor* and *S. stoloniferum*, a multiple comparisons were conducted by the least significant difference (LSD) method. The “Multiple–regression” package in the R 4.4.2 statistical analysis software “www.r-project.org (accessed on 17 August 2024)” was used to perform stepwise regression analysis on the functional traits of the two plants and the temperature factors, respectively, to select the main temperature factors affecting the traits of *Z. caduciflor* and *S. stoloniferum*. Redundancy analysis (RDA) was conducted using Canoco 5 software “www.canoco5.com (accessed on 28 August 2024)” to examine the relationships between the traits of *Z. caduciflor* and *S. stoloniferum* and the temperature factors. Pearson correlation analysis was performed using SPSS software to further detect the correlations between “traits and traits” and “traits and temperature factors” in the two plants. The significance levels for all data analyses in this study are defined as follows: *p* < 0.05 indicates significance, *p* < 0.01 indicates highly significant, and *p* < 0.001 indicates extremely significant. The figures in this study were created using Origin 2018 and Adobe Photoshop 2020 software.

## 4. Discussion

*Z. caduciflora* and *S. stoloniferum* are both typical emergent plants in the Napahai lakeshore zone. Simulated warming significantly reduced the net photosynthetic rate and stomatal conductance of *Z. caduciflora* (Figure 1), but these photosynthetic parameters of *S. stoloniferum* did not show significant changes (Figure 1). This indicates that *Z. caduciflora* is more sensitive to simulated warming, with a 2 °C increase already exceeding its optimal growth temperature, while *S. stoloniferum* can maintain a stable photosynthetic capacity under simulated warming conditions. Previous studies on aquatic plants such as *Typha orientalis* and *Phragmites australis* in high–altitude areas also showed that warming would decrease plant photosynthetic productivity [33,34], which is consistent with the trend observed in *Z. caduciflora*, suggesting that the decline in the photosynthetic productivity of large aquatic plants in plateau regions is a common phenomenon under the background of climate warming. When environmental temperatures exceed the optimal temperature for plant growth, the net photosynthetic rate will decline [35]. Even though an increase in temperature can enhance the actual photosynthetic rate of plants, the increase in photorespiration and plant respiration under high temperatures can also lead to a decrease in the net photosynthetic rate [36]. During the carbon fixation process, ribulose–1,5–bisphosphate carboxylase/oxygenase (Rubisco) can promote photosynthetic efficiency. The amount of active Rubisco is limited by the activity of Rubisco activase, which is inhibited by environmental conditions such as increased temperature, shading, or elevated CO_2_ concentrations [37,38]. In addition to this direct effect, warming can also affect the activity of Rubisco activase indirectly by reducing the rate of electron transport, redox potential, and the amount of nucleoside triphosphates, thereby decreasing photosynthetic efficiency and other processes related to photosynthesis [36,37]. Therefore, increasing the thermal stability of plants will be more conducive to maintaining higher net photosynthetic rates in plants [39].

Although the photosynthetic parameters of *Z. caduciflora* and *S. stoloniferum* respond differently to simulated warming, the biomass of both species shows a declining trend under simulated warming conditions, with decreases in the aboveground, belowground, and total biomass under warming conditions (Figure 1), indicating that high temperatures inhibit the biomass accumulation of both *Z. caduciflora* and *S. stoloniferum*. The aboveground part is the plant’s nutritional tissue, while the belowground part is related to the plant’s clonal reproduction; therefore, warming significantly reduces the nutritional growth and offspring clonal reproduction capabilities of both plants. Generally, an increase in temperature can extend the growing season by advancing the arrival of spring and delaying the onset of winter; however, for many plants, moderate warming can shorten the time required for plant development [40,41]. For instance, appropriate warming before plants reach maturity can shorten the leaf construction time, thereby shortening the plant development process [40,41]. This rapid development process in plants will reduce the total carbon fixation, which in turn decreases the energy investment in nutritional and reproductive structures [42]. Temperature increases beyond the species’ optimal temperature may affect the genetic potential of plants in terms of adaptation and biomass, thereby endangering the species’ survival capabilities [42]. Some studies have found that the regulatory capacity of the aboveground parts is stronger than that of the belowground parts; under resource–limited conditions, plants generally increase the proportion of underground biomass to ensure the reproductive capacity of offspring [33,43]. Compared to these plants, under the background of long–term climate warming, *Z. caduciflora* and *S. stoloniferum* may have reduced investment in offspring, leading to lower competitive ability, which is not conducive to the future development of the plants.

Consistent with the decline in photosynthetic capacity, the aboveground nitrogen content of *Z. caduciflora* and the nitrogen content of its underground buds were significantly reduced, and the aboveground nitrogen content and root nitrogen content of *S. stoloniferum* were also significantly decreased (Figure 2). Nitrogen is a structural element of photosynthetic proteins, especially the essential protein for photosynthesis–the Rubisco enzyme–and it is also the raw material for all protein metabolism and amino acid synthesis within plants. Global leaf economic spectra and wetland plant economic spectra have both demonstrated a significant positive correlation between the net photosynthetic rate and nitrogen and phosphorus content [44,45]. Higher photosynthetic rates, higher nitrogen and phosphorus content, and lower specific leaf weight are typical characteristics of resource–acquisitive plants, while lower net photosynthetic rates and nitrogen and phosphorus content, along with higher specific leaf weight, are typical characteristics of resource–conservative plants [26]. Wetland plants are generally clustered towards the “low investment–rapid return” end of resource acquisitive strategies [44]. The significant reduction in the net photosynthetic rate and nitrogen content of *Z. caduciflora* under warming conditions indicates that as temperatures rise, its resource utilization type is gradually shifting towards a resource–conservative strategy. Warming reduces the supply of nitrogen by decreasing the activity and quantity of enzymatic and other material activities, thereby reducing the net photosynthetic rate. Therefore, the deficiency of mineral nutrients under warming conditions is one of the reasons for the decline in the photosynthetic productivity of *Z. caduciflora*. *S. stoloniferum*’s response strategy differs from that of *Z. caduciflora*. In terms of the decrease in nitrogen content, the response strategies of *Z. caduciflora* and *S. stoloniferum* are consistent. However, regarding phosphorus, the phosphorus content of *Z. caduciflora* remains unchanged under all temperature conditions, while the phosphorus content of all underground parts of *S. stoloniferum*, including interval, fibrous roots, main roots, and underground buds, is significantly higher under warming conditions compared to the control group (Figure 2), indicating that simulated warming promotes the absorption of phosphorus in the underground parts of *S. stoloniferum*. Phosphorus is an important component of nucleic acids, lipid membranes, and the biological energy molecule ATP [46]. Under natural conditions, increasing plants’ intake of phosphorus can, to some extent, mitigate the impact of nitrogen deficiency on plant photosynthetic physiology and effectively ensure normal plant growth in the short term [46]. This may be one of the important reasons why *S. stoloniferum* can maintain stable photosynthetic rates under warming conditions.

The mean annual temperature (MAT) and the annual maximum temperature (max) are the main temperature factors affecting the functional traits of *Z. caduciflora* and *S. stoloniferum* (Table 3 and Table 4). The MAT reflects the average temperature level during the plant growth process and is directly related to the length of the growing season and the effective accumulated temperature during the growing period, while the max represents the extreme high temperatures in the plant growth environment [25,33,47]. These two temperature factors are mainly negatively correlated with the photosynthetic parameters, biomass of various parts, and nitrogen content in the aboveground parts and underground buds of *Z. caduciflora* and *S. stoloniferum* (Figure 3, Table 3), indicating that the increase in the MAT and max leads to a decrease in water vapor exchange capacity and nitrogen content in the aboveground parts and underground buds, resulting in insufficient photosynthetic raw materials and nutrient supply, a decline in photosynthetic productivity, and subsequently reduced biomass accumulation. Generally speaking, sustained high temperatures can reduce the photosynthetic efficiency of plants, shorten their life cycles, and lead to a decrease in productivity [48]. Under extreme high–temperature conditions, plants are further directly inhibited at the physiological level, and their metabolism is affected. Extreme high–temperature stress can impact the structure of chloroplasts and the thermal stability of photosynthetic system components, reducing the activity of Rubisco, the content of photosynthetic pigments, and the capacity for carbon fixation [49]. For plants, the survival of chloroplasts, which are essential for photosynthesis, has a certain temperature range. Excessively high temperatures can damage the thylakoid membrane, inhibit the activity of membrane–associated electron carriers and enzymes, and promote too rapid an electron transfer rate of chlorophyll a, leading to a decline or even interruption of the activity of heat–sensitive Photosystem II (PS II), causing irreversible tissue damage [50,51]. Plants possess a variety of mechanisms that help improve their heat tolerance, allowing them to endure high temperatures and prevent damage from excessively high leaf temperatures. These mechanisms include heat conduction, heat convection, and transpirational cooling [52,53]. Under water–limited conditions, plant species with low stomatal conductance due to isohydric behavior or inherently low transpiration capacity will open their stomata and increase stomatal conductance under high temperatures; however, under well–watered conditions, high temperatures will prompt plants to close their stomata, reducing transpiration rates and stomatal conductance [54]. The decrease in the transpiration rate and stomatal conductance reduces the capacity for gas exchange, leading to a decline in photosynthesis due to insufficient water and CO_2_ and, consequently, a reduction in biomass accumulation. There is, therefore, a balance between plant physiology and leaf energy [55]. For wetland plants, the limiting effect of water availability is relatively weak, so reducing tissue damage and ensuring the normal operation of the photosynthetic system under extreme high temperatures may be the main reason for the decrease in stomatal conductance and the transpiration rate in aquatic plants under high temperatures.

Nutrient supply, especially the supply of nitrogen and phosphorus, is an important aspect of photosynthesis in plants. Insufficient nutrients can directly lead to a lack of energy and material supply for the photosynthetic process, thereby reducing photosynthetic efficiency [56]. The relationship between the nitrogen content in the aboveground parts and underground buds of *Z. caduciflora* and *S. stoloniferum* and temperature factors is the strongest, while the relationship between carbon and phosphorus content and these two temperature factors is mostly insignificant. This indicates that under warming conditions, these two plants are more sensitive to nitrogen limitations and relatively less sensitive to carbon and phosphorus limitations. It also indicates that the aboveground parts and underground buds are key structures in determining the photosynthetic productivity of plants in response to warming. Studies on the absorption of nitrogen and phosphorus elements of short–eared rabbit grass on the Qinghai–Tibet Plateau under warming conditions also found that warming has a greater impact on nitrogen absorption and a less noticeable impact on phosphorus absorption [57]. The results of this study are in line with the conclusions of global pattern research on the terrestrial nitrogen and phosphorus limitation, which is that high–altitude and high–latitude ecosystems typically show a strong nitrogen limitation and have a lower demand for external phosphorus [58]. The aboveground parts, especially leaves, are the main structures for plants to carry out photosynthesis. According to the global plant leaf economic spectrum and wetland plant leaf economic spectrum, the nitrogen content in the aboveground parts is directly related to plant photosynthesis [26,44], and a decrease in its content directly leads to a decrease in the photosynthetic rate. Many studies have shown that climate change, especially temperature changes, can alter the content and combination ratios of elements within plants, thereby leading to changes in ecosystem functions [59,60,61]. However, the impact of climate change on plant elemental content is indirect [59]. For example, extreme high–temperature stress can lead to the production of free radicals within plants, affecting the recycling process of nitrogen, with incomplete nitrogen absorption being an important reason for the decline in plant nitrogen content [10]. The underground bud is the main structure for the clonal reproduction of aquatic plants. Under high temperatures, the decline in nitrogen accumulation and the reduction in underground bud biomass both indicate that the reproductive capacity of aquatic plants decreases under the influence of high temperatures. Therefore, the 2 °C and 4 °C warming in this study is both high–temperature stress for *Z. caduciflora* and *S. stoloniferum*. Against the backdrop of climate warming, the survival, reproduction, and development of *Z. caduciflora* and *S. stoloniferum* will face challenges, and the distribution pattern of plants and the stability of ecosystems will also be affected.

## 5. Conclusions

This study investigated the effects of warming on the photosynthetic traits, biomass, and elemental contents of two typical emergent plants (*Z. caduciflora* and *S. stoloniferum*) in the Napahai lakeshore zone through a simulated warming experiment. The results showed that simulated warming significantly reduced the net photosynthetic rate and stomatal conductance of *Z. caduciflora*, but had no significant impact on the photosynthetic parameters of *S. stoloniferum*. This indicates that *Z. caduciflora* is more sensitive to warming, while *S. stoloniferum* can maintain stable photosynthetic capacity under warming conditions. The biomass of both plants showed a declining trend under warming conditions, indicating that high temperatures inhibit biomass accumulation in both species. Warming not only shortened the growing season of the plants but also reduced their energy investment in vegetative growth and clonal reproduction. Additionally, warming led to a significant decrease in nitrogen content in both plants, particularly in the aboveground parts and underground buds of *Z. caduciflora*, which further limited its photosynthetic productivity. In contrast, *S. stoloniferum* alleviated the impact of nitrogen limitation by increasing phosphorus content in its underground parts, thus maintaining a stable photosynthetic rate. The mean annual temperature (MAT) and the annual maximum temperature (max) are the primary temperature factors influencing the traits of these two plants. The increase in the MAT and max led to a decrease in the plants’ water vapor exchange capacity and nitrogen content, thereby affecting photosynthetic productivity and biomass accumulation. Moreover, under high temperature stress, plants regulate their stomatal conductance and transpiration rates to maintain physiological balance, but this also restricts gas exchange capacity, further reducing photosynthetic efficiency.

## Figures and Tables

**Figure 1 plants-14-01414-f001:**
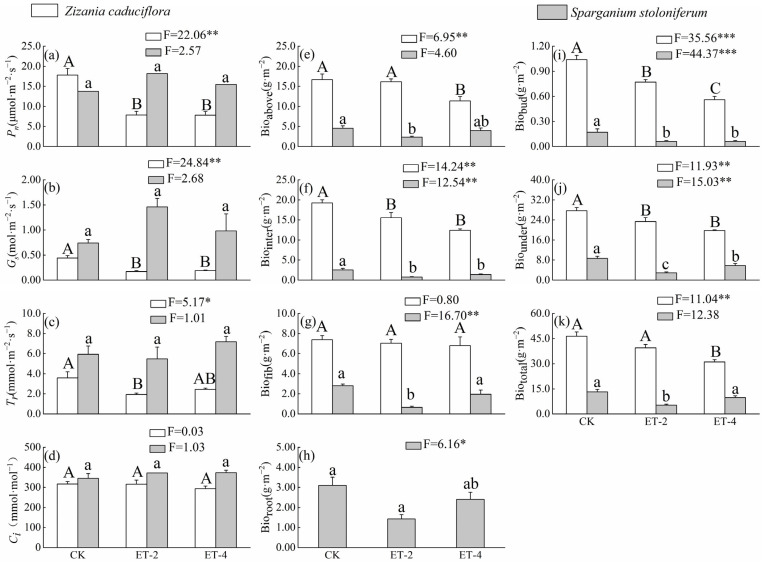
Differences in photosynthetic traits and biomass of *Z. caduciflor* and *S. stoloniferum* among three different growth temperature groups. In this context, the underground part of the *Z. caduciflor* does not include the roots. Different lowercase letters indicate significant differences of plant traits in *S. stoloniferum* at the 0.05 level, while different uppercase letters indicate significant differences of plant traits in *Z. caduciflor* at the 0.05 level (*p* < 0.05). (**a**) *P_n_*, net photosynthetic rate; (**b**) *G_s_*, stomatal conductance; (**c**) *T_r_*, transpiration rate; (**d**) *C_i_*, intercellular CO_2_ concentration; (**e**) Bio_above_, aboveground biomass; (**f**) Bio_inter_, interval biomass; (**g**) Bio_fib_, fibrous root biomass; (**h**) Bio_root_, root biomass; (**i**) Bio_bud_, bud biomass; (**j**) Bio_under_, underground biomass; (**k**) Bio_total_, total biomass. CK, ambient temperature (control group); ET–2, (2.0 ± 0.5) °C warming; ET–4, (4.0 ± 0.5) °C warming. * *p* < 0.05; ** *p* < 0.01; *** *p* < 0.001.

**Figure 2 plants-14-01414-f002:**
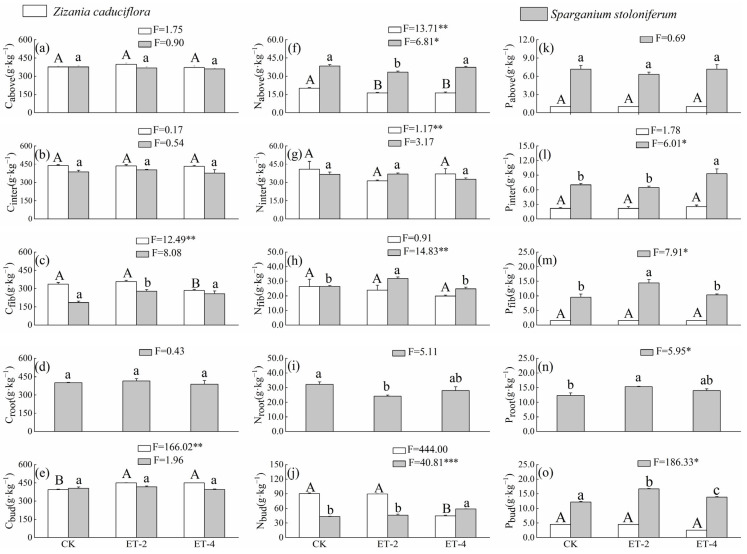
Differences in elemental mass fractions of *Z. caduciflora* and *S. stoloniferum* among three different growth temperature groups. In this context, the underground part of the *Z. caduciflor* does not include the roots. Different lowercase letters indicate significant differences in plant traits in *S.stoloniferum* at the 0.05 level, while different uppercase letters indicate significant differences in plant traits in *Z. caduciflor* at the 0.05 level (*p* < 0.05). (**a**) C_above_, aboveground carbon comtent; (**b**) C_inter_, interval carbon comtent; (**c**) C_fib_, fibrous root carbon comtent; (**d**) C_root_, root carbon comtent; (**e**) C_bud_, bud carbon comtent; (**f**) N_above_, aboveground nitrogen comtent; (**g**) N_inter_, interval nitrogen comtent; (**h**) N_fib_, fibrous root nitrogen comtent; (**i**)N_root_, root nitrogen comtent; (**j**) N_bud_, bud nitrogen comtent; (**k**) P_above_, aboveground phosphorus comtent; (**l**) P_inter_, interval phosphorus comtent; (**m**) P_fib_, fibrous root phosphorus comtent; (**n**) P_root_, root phosphorus comtent; (**o**) P_bud_, bud phosphorus comtent. CK, ambient temperature (control group); ET–2, (2.0 ± 0.5) °C warming; ET–4, (4.0 ± 0.5) °C warming. * *p* < 0.05; ** *p* < 0.01; *** *p* < 0.001.

**Figure 3 plants-14-01414-f003:**
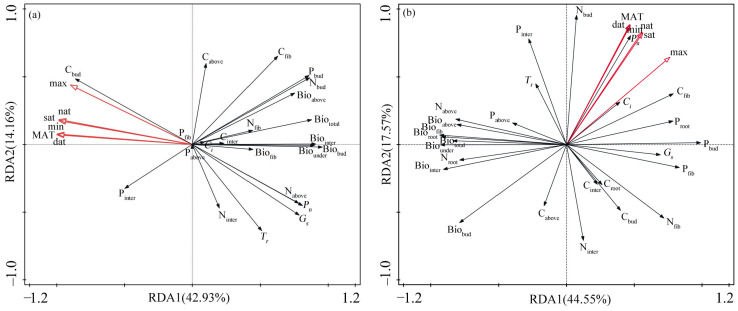
Association between functional traits and temperature factors of two plants ((**a**) *Z. caduciflora*; (**b**) *S. stoloniferum*) based on RDA analysis. In this context, the underground part of the *Z. caduciflor* does not include the roots. Abbreviations for traits in the table are as shown in Table 5. MAT: mean annual temperature, max: annual maximum temperature, min: annual minimum temperature, sat: seasonal average temperature, dat: daytime accumulated temperature, nat: nighttime accumulated temperature.

**Figure 4 plants-14-01414-f004:**
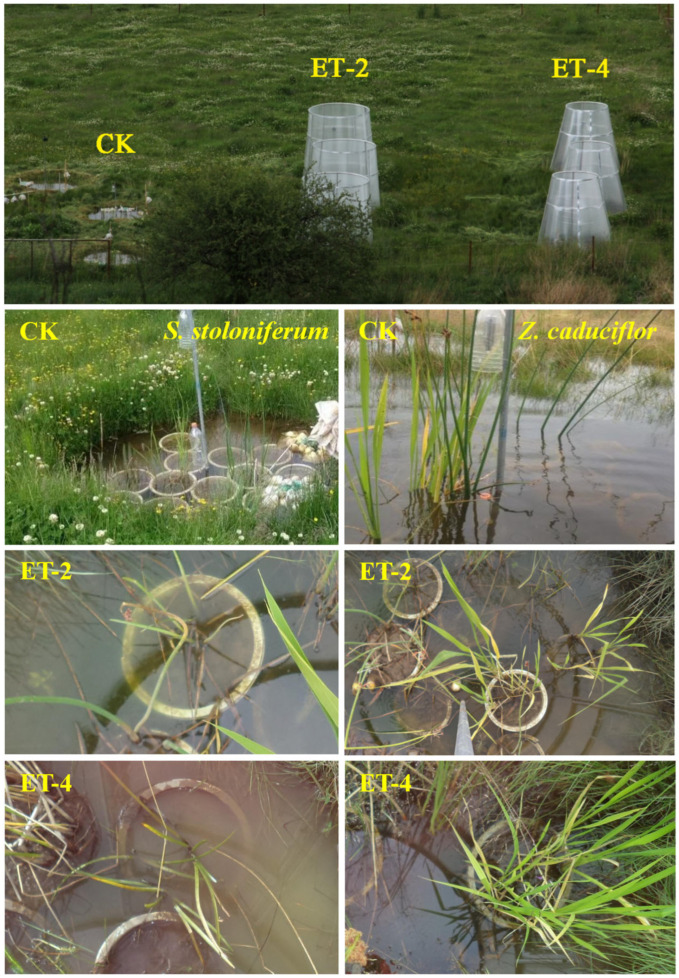
Open–top chambers in Napahai wetland and the growth status of the two plant species in the chambers. CK, ambient temperature (control group); ET–2, (2.0 ± 0.5) °C warming; ET–4, (4.0 ± 0.5) °C warming.

**Table 1 plants-14-01414-t001:** Stepwise regression models based on the traits of *Z. caduciflora* and temperature factors.

Traits	Variable	Coefficient	*R* ^2^	*p*	Traits	Variable	Coefficient	*R* ^2^	*p*
*P_n_*	max	−1.05	0.89	0.000	C_above_	MAT	−19.90	0.37	0.253
	Constant	57.19				max	7.16		
						Constant	282.04		
*G_s_*	max	−0.03	0.88	0.000	C_inter_	none			
	Constant	1.46							
*T_r_*	max	−0.14	0.55	0.022	C_fib_	MAT	−50.66	0.81	0.007
	Constant	9.01				max	14.24		
						Constant	247.12		
*C_i_*	none				C_bud_	max	5.76	0.98	<0.001
						Constant	177.19		
Bio_above_	MAT	−3.27	0.69	0.027	N_above_	max	−0.42	0.82	0.001
	max	0.71				Constant	35.98		
	Constant	18.75							
Bio_inter_	MAT	−1.81	0.82	0.001	N_inter_	none			
	Constant	35.34							
Bio_fib_	MAT	−0.16	0.21	0.215	N_fib_	MAT	−1.69	0.22	0.205
	Constant	8.77				Constant	41.66		
Bio_bud_	MAT	−0.13	0.92	3.974 × 10^−5^	N_bud_	MAT	−29.02	0.99	<0.001
	Constant	2.18				max	6.73		
						Constant	93.03		
Bio_under_	MAT	−2.10	0.80	0.001	P_above_	none			
	Constant	46.31							
Bio_total_	MAT	−3.45	0.74	0.003	P_inter_	MAT	0.17	0.25	0.171
	Constant	75.57				Constant	0.59		
					P_fib_	none			
					P_bud_	none			

Abbreviations for traits in the table are as shown in Table 5. MAT, mean annual temperature; max, annual maximum temperature.

**Table 2 plants-14-01414-t002:** Stepwise regression models based on the traits of *S. stoloniferum* and temperature factors.

Traits	Variable	Coefficient	*R* ^2^	*p*	Traits	Variable	Coefficient	*R* ^2^	*p*
*P_n_*	MAT	−2.09	0.46	0.156	C_fib_	max	8.64	0.67	0.007
	max	0.97				Constant	−140.33		
	Constant	−4.54							
*G_s_*	MAT	−0.37	0.47	0.148	C_root_	none			
	max	0.16							
	Constant	−2.24							
*T_r_*	none				C_bud_	MAT	−15.74	0.39	0.222
						max	4.99		
						Constant	356.57	0.81	0.007
*C_i_*	max	2.82	0.26	0.165	N_above_	MAT	3.01	0.69	0.029
	Constant	239.34				Max	−1.25		
						Constant	59.05		
Bio_above_	MAT	1.22	0.61	0.062	N_inter_	MAT	−2.90	0.51	0.115
	max	−0.53				max	0.70		
	Constant	13.62				Constant	35.700		
Bio_inter_	MAT	0.51	0.81	0.007	N_fib_	MAT	−5.23	0.83	0.005
	max	−0.31				max	1.84		
	Constant	9.82				Constant	3.29		
Bio_fib_	MAT	1.00	0.85	0.004	N_root_	MAT	3.04	0.63	0.051
	max	−0.47				max	−1.61		
	Constant	11.73				Constant	66.04		
Bio_root_	MAT	0.75	0.67	0.035	N_bud_	MAT	8.64	0.93	0.000
	max	−0.36				max	−1.71		
	Constant	9.98				Constant	31.35		
Bio_bud_	max	−0.01	0.94	2.092 × 10^−5^	P_above_	none			
	Constant	0.63							
Bio_under_	MAT	2.27	0.834	0.005	P_inter_	MAT	2.01	0.67	0.037
	max	−1.16				max	−0.54		
	Constant	32.24				Constant	9.44		
Bio_total_	MAT	3.49	0.81	0.007	P_fib_	MAT	−3.07	0.725	0.021
	max	−1.68				max	1.26		
	Constant	45.87				Constant	−10.68		
C_above_	MAT	−4.20	0.23	0.195	P_root_	MAT	−1.06	0.67	0.038
	Constant	413.78				max	0.58		
						Constant	−0.04		
C_inter_	none				P_bud_	MAT	−2.17	0.98	<0.001
						max	1.00		
						Constant	−6.44		

Abbreviations for traits in the table are as shown in Table 5. MAT, mean annual temperature; max, annual maximum temperature.

## Data Availability

The data that support the findings of this study are available from the corresponding author upon reasonable request. Please contact the author at sm0510215@163.com.

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
