# Peer review of "Simulated Warming Reduces Biomass Accumulation in Zizania caduciflor and Sparganium stoloniferum"

_plants, 2025, doi:10.3390/plants14101414_

Round 1
Reviewer 1 Report
Comments and Suggestions for Authors
Global warming, a paramount environmental challenge, significantly impacts the structure and function of alpine wetland ecosystems. This manuscript investigated the responses of the representative alpine wetland plants, Zizania caduciflor and Sparganium stoloniferum, to simulated warming via an open-top chamber experiment. The findings demonstrated that elevated temperatures reduce photosynthetic productivity and biomass accumulation in these alpine wetland plants. This study provided further insights into the response strategies of plants in the alpine wetlands of Northwest Yunnan to climate warming, which will contribute to more effective scientific responses to climate change. Here are some comments:
- The abstract requires streamlining to concisely present the key findings.
- Specify the multiple comparison method employed in the statistical analyses.
- Provide a rationale for the selection of Zizania caduciflor and Sparganium stoloniferum as study subjects.
- Clarify the meaning of the uppercase letters used in Figures 2 and 3.
- Ensure the accurate reporting of results. For instance, in Section 3.1, there was no significant difference in Tr between the ET-4 and CK treatments of Z. caduciflor. The Bioabove, Biounder, and Biototal in S. stoloniferum does not show a significant downward trend with increasing temperature as described. A thorough review of the results analysis throughout the manuscript is recommended.
- Add photos of plants under various treatments in the manuscript to visually display their growth status.
- Consider presenting the data from Tables 3 and 4 in a visual figures.
- Define the significance of the *** in Table 5. Additionally, incorporate the significance level (P < 0.01) in the Section 2.4 Data analysis.
- Incorporate a dedicated conclusion section.
- Identify the authors responsible for the initial manuscript drafting in the author contributions section.
- It is necessary to italicize the genus and species names in the references.
Author Response
1. Summary |
|
|
Thank you very much for taking the time to review this manuscript. We now have thoroughly considered and incorporated all comments given by the referee, and substantially improved the manuscript. Please find the detailed responses below and the corresponding corrections highlighted in the re-submitted files. |
||
2. Questions for General Evaluation |
Reviewer’s Evaluation |
Response and Revisions |
Does the introduction provide sufficient background and include all relevant references? |
Yes |
Thank you for your evaluation |
Is the research design appropriate? |
Yes |
Thank you for your evaluation |
Are the methods adequately described? |
Can be improved |
The methods section has been supplemented and revised in the revised manuscript. |
Are the results clearly presented? |
Must be improved |
The results have been thoroughly verified and revised in the revised manuscript. |
Are the conclusions supported by the results? |
Can be improved |
A dedicated conclusion section has been added after the discussion section in the revised manuscript. |
3. Point-by-point response to Comments and Suggestions for Authors |
||
General Evaluation: [Global warming, a paramount environmental challenge, significantly impacts the structure and function of alpine wetland ecosystems. This manuscript investigated the responses of the representative alpine wetland plants, Zizania caduciflor and Sparganium stoloniferum, to simulated warming via an open-top chamber experiment. The findings demonstrated that elevated temperatures reduce photosynthetic productivity and biomass accumulation in these alpine wetland plants. This study provided further insights into the response strategies of plants in the alpine wetlands of Northwest Yunnan to climate warming, which will contribute to more effective scientific responses to climate change.] |
||
Comments 1: [The abstract requires streamlining to concisely present the key findings.] |
||
Response 1: We are very grateful for the suggestions provided by the reviewer. We have streamlined the description of the results in the abstract, retaining only the most critical findings. Additionally, we have improved a few expressions.——Lines 15 to 26. Comments 2: [Specify the multiple comparison method employed in the statistical analyses.] Response 2: Thank you for the reviewer's reminder. We have added the method of multiple comparisons (least significant difference, LSD) in the data analysis section and included the URLs of the statistical analysis software we used. ——Lines 233 to 238. Comments 3: [Provide a rationale for the selection of Zizania caduciflor and Sparganium stoloniferum as study subjects.] Response 3: We appreciate the suggestion from the reviewer. In the last paragraph of the Introduction, we have added the ecological roles of the two plants in Northwest Yunnan as the original reasons for choosing these plants for our study. ——Lines 85 to 89. Comments 4: [Clarify the meaning of the uppercase letters used in Figures 2 and 3.] Response 4: The information has already been supplemented in the revised manuscript.——Lines 283 to 284, 322 to 329. Comments 5: [Ensure the accurate reporting of results. For instance, in Section 3.1, there was no significant difference in Tr between the ET-4 and CK treatments of Z. caduciflor. The Bioabove, Biounder, and Biototal in S. stoloniferum does not show a significant downward trend with increasing temperature as described. A thorough review of the results analysis throughout the manuscript is recommended.] Response: We apologize for this mistake. In the revised manuscript, we have thoroughly verified and amended the descriptions in the results section based on the figures and tables.——Lines 252 to 275. Comments 6: [Add photos of plants under various treatments in the manuscript to visually display their growth status.] Response: We are grateful for the reviewer's suggestions. Photos of the plants under the three growth conditions have been added to Figure 1. Since we did not specifically plan to take photos of the plant growth status during the study, the number of available photos is limited and the quality is not very good. We apologize for this inconvenience. The reviewer's suggestions have also reminded us to take and retain more photos in future research.——Line 167. Comments 7: [Consider presenting the data from Tables 3 and 4 in a visual figure.] Response: Thank you for the reviewer' suggestion. However, we are very sorry that, despite our careful attempts, we still cannot convert these two tables into more visually appealing figures. Tables 3 and 4 present the main temperature factors closely related to the traits of the two plants, as selected by the stepwise regression model. The tables show the parameters selected by the model, the model's coefficient of determination (R²), and the significance level (P). Comments 8: [Define the significance of the *** in Table 5. Additionally, incorporate the significance level (P < 0.01) in the Section 2.4 Data analysis.] Response: Done, we have already defined the meaning of "***" in the footnote of Table 5, and supplemented the significance level of the statistical analysis in Section 2.4 Data Analysis. ——Lines 246 to 248 and line 408. Comments 9: [Incorporate a dedicated conclusion section.] Response: Thank you to the reviewer for his/her suggestion. A dedicated conclusion section has been incorporated into the revised manuscript.——Lines 573 to 597. Comments 10: [Identify the authors responsible for the initial manuscript drafting in the author contributions section.] Response: This information has been supplemented in the revised manuscript.——Line 599. Comments 11: [It is necessary to italicize the genus and species names in the references.] Response: We appreciate the reviewer for pointing out this error. We have carefully checked all the references and changed the Latin names of the species to italics throughout the manuscript. |
||
4. Response to Comments on the Quality of English Language |
||
Point: The English is fine and does not require any improvement. |
||
Response: Thank you for your evaluation.
|

Reviewer 2 Report
Comments and Suggestions for Authors
[Plants] Manuscript ID: plants-3593785
Simulated warming reduces biomass accumulation in Zizania caduciflor and Sparganium stoloniferum
1. The authors have highlighted a very important topic of climate change and its impact on ecosystem productivity.
2. This study investigates how climate change—particularly warming—impacts alpine wetland ecosystems using open-top chamber (OTC) warming experiments at the Napahai alpine wetland in Yunnan, China. The focus was on two typical wetland plant species: Zizania caduciflora and Sparganium stoloniferum.
3. The authors have used detailed inferential statistical analysis to explore their results.
4. The results provide detailed insights into the physiological and morphological responses of two studied species under these stimulated warming conditions.
5. Z. caduciflora: Most functional traits (e.g., biomass, nitrogen content) were negatively correlated with mean annual temperature (MAT) and annual max temperature (max).
6. S. stoloniferum: Biomass and nitrogen content had negative correlations with max, while carbon and phosphorus content in roots showed positive correlations.
7. The study aligns with other findings from high-altitude regions, suggesting a broader pattern of climate change reducing photosynthetic efficiency and biomass.
8. Long-term warming may lead to changes in vegetation composition and declining productivity in alpine wetlands.
9. This research enhances understanding of plant response strategies to global warming in alpine wetlands and supports climate adaptation and conservation planning for fragile high-altitude ecosystems.
10. The paper has 23% similarity (checked through Turnitin) with other published work.
However, there are a few points that need to be addressed by the authors. The points are listed below.
Points to be addressed:
a. Line 122: Please rephrase this sentence – “with an accumulated temperature of 10°C activity of 1392.8°C” to this “Using a base temperature of 10°C, the accumulated temperature (or thermal time or Growing Degree Days, GDD) reached 1392.8°C.
b. Line 128: Please follow one type of reference style as suggested by journal as here the authors have mentioned (Xiao et al., 2008), whereas in all other references they are using numbers e.g., [1] [2] etc.
Author Response
1. Summary |
|
|
Thank you very much for taking the time to review this manuscript. We now have thoroughly considered and incorporated all comments given by the referee, and substantially improved the manuscript. Please find the detailed responses below and the corresponding corrections highlighted in the re-submitted files.
|
||
2. Questions for General Evaluation |
Reviewer’s Evaluation |
Response and Revisions |
Does the introduction provide sufficient background and include all relevant references? |
Yes |
Thank you for your evaluation |
Is the research design appropriate? |
Yes |
Thank you for your evaluation |
Are the methods adequately described? |
Yes |
Thank you for your evaluation |
Are the results clearly presented? |
Yes |
Thank you for your evaluation |
Are the conclusions supported by the results?
|
Yes |
Thank you for your evaluation |
3. Point-by-point response to Comments and Suggestions for Authors |
||
General Evaluation: [ 1. The authors have highlighted a very important topic of climate change and its impact on ecosystem productivity. 2. This study investigates how climate change—particularly warming—impacts alpine wetland ecosystems using open-top chamber (OTC) warming experiments at the Napahai alpine wetland in Yunnan, China. The focus was on two typical wetland plant species: Zizania caduciflora and Sparganium stoloniferum. 3. The authors have used detailed inferential statistical analysis to explore their results. 4. The results provide detailed insights into the physiological and morphological responses of two studied species under these stimulated warming conditions. 5. Z. caduciflora: Most functional traits (e.g., biomass, nitrogen content) were negatively correlated with mean annual temperature (MAT) and annual max temperature (max). 6. S. stoloniferum: Biomass and nitrogen content had negative correlations with max, while carbon and phosphorus content in roots showed positive correlations. 7. The study aligns with other findings from high-altitude regions, suggesting a broader pattern of climate change reducing photosynthetic efficiency and biomass. 8. Long-term warming may lead to changes in vegetation composition and declining productivity in alpine wetlands. 9. This research enhances understanding of plant response strategies to global warming in alpine wetlands and supports climate adaptation and conservation planning for fragile high-altitude ecosystems. 10. The paper has 23% similarity (checked through Turnitin) with other published work. ] |
||
Points to be addressed: |
||
Comments 1: [ Line 122: Please rephrase this sentence – “with an accumulated temperature of 10°C activity of 1392.8°C” to this “Using a base temperature of 10°C, the accumulated temperature (or thermal time or Growing Degree Days, GDD) reached 1392.8°C.] |
||
Response 1: Thank you to the reviewer for pointing out this error. It has been corrected in the revised manuscript. ——Lines 124 to 126. Comments 2: [ Line 128: Please follow one type of reference style as suggested by journal as here the authors have mentioned (Xiao et al., 2008), whereas in all other references they are using numbers e.g., [1] [2] etc.] Response 2: Thank you to the reviewer for pointing out this error. It has been corrected in the revised manuscript. ——Line 128.
|
||
4. Response to Comments on the Quality of English Language |
||
Point: The English is fine and does not require any improvement. |
||
Response: Thank you for your evaluation.
|

Round 2
Reviewer 1 Report
Comments and Suggestions for Authors
The authors have responded to the comments and carefully revised the manuscript.
Author Response
Response to Reviewer 1 Comments |
||
1. Summary |
|
|
Thank you very much for taking the time to review this manuscript. We now have thoroughly considered and incorporated all comments given by the referee, and substantially improved the manuscript. Please find the detailed responses below and the corresponding corrections highlighted in the re-submitted files. |
||
2. Questions for General Evaluation |
Reviewer’s Evaluation |
Response and Revisions |
Does the introduction provide sufficient background and include all relevant references? |
Yes |
Thank you for your evaluation |
Is the research design appropriate? |
Yes |
Thank you for your evaluation |
Are the methods adequately described? |
Yes |
Thank you for your evaluation |
Are the results clearly presented? |
Yes |
Thank you for your evaluation |
Are the conclusions supported by the results? |
Yes |
Thank you for your evaluation |
3. Point-by-point response to Comments and Suggestions for Authors |
||
Comments and Suggestions for Authors: [The authors have responded to the comments and carefully revised the manuscript.] |
||
Response : Thank you for your evaluation. |
